# Phase Separation within a Thin Layer of Polymer Solution as Prompt Technique to Predict Membrane Morphology and Transport Properties

**DOI:** 10.3390/polym12122785

**Published:** 2020-11-25

**Authors:** Tatiana Anokhina, Ilya Borisov, Alexey Yushkin, Gleb Vaganov, Andrey Didenko, Alexey Volkov

**Affiliations:** 1A.V. Topchiev Institute of Petrochemical Synthesis, Russian Academy of Sciences, Leninsky pr., 29, 119991 Moscow, Russia; tsanokhina@ips.ac.ru (T.A.); halex@ips.ac.ru (A.Y.); avolkov@ips.ac.ru (A.V.); 2Institute of Macromolecular Compounds, Russian Academy of Sciences, Bolshoy pr. V.O., 31, 199004 St. Petersburg, Russia; glebvaganov@mail.ru (G.V.); vanilin72@yandex.ru (A.D.)

**Keywords:** polyimides, polyamic acid, flat-sheet membrane, non-solvent induced phase separation, casting solution, kinetics of solution precipitation

## Abstract

In this work, the precipitation of a thin layer of a polymer solution was proposed to imitate the process of asymmetric membrane formation by a non-solvent induced phase separation (NIPS) technique. The phase inversion within the thin (<500 μm) and bulk (~2 cm) layer of polyamic-acid (PAA) in N-methyl-2-pyrrolidone (NMP) by using water as non-solvent was considered. It was shown that polymer films formed within the “limited” layer of polymer solution showed a good agreement with the morphology of corresponded asymmetric flat-sheet membranes even in the case of three-component casting solution (PAA/NMP/EtOH). At the same time, the polymer films formed on the interface of two bulk phases (“infinite” regime) did not fully correspond to the membrane structure. It was shown that up to 50% of NMP solvent in PAA solution can be replaced by ethanol, which can have a renewable origin. By changing the ethanol content in the casting solution, the average size of transport pores can be varied in the range of 12–80 nm, and the liquid permeance from 16.6 up to 207 kg/m^2^∙h∙bar. To summarize, the precipitation of polymer solution within the thin layer can be considered a prompt technique and a powerful tool for fast screening and optimization of the complex composition of casting solutions using its small quantity. Furthermore, the prediction of membrane morphology can be done without casting the membrane, further post-treatment procedures, and scanning electron microscopy (SEM) analysis.

## 1. Introduction

Aromatic polyimides (PI) are materials with high mechanical strength and chemical stability due to the rigid structure of the polymer backbone and strong intramolecular and intermolecular interactions [1,2,3,4]. However, their insolubility in organic solvents complicates their treatment and the process of preparation of membranes based on them [5,6]. For aromatic polyimides insoluble in organic solvents, Schumann and Strathmann first proposed and patented the process which includes the formation of ultrafiltration and reverse osmosis flat membranes from solutions of a PAA prepolymer with its subsequent thermal or chemical imidization to obtain PI membranes [7].

The difficulty of PAA-based membrane formation by NIPS is often associated with high viscosity of the polymer solution due to the presence of a large number of carboxyl groups that form intermolecular hydrogen bonds. This in turn leads to a low deposition rate in the polymer solution. Therefore, the process of choosing the optimal composition of the casting solution for the PAA-based membranes is very challenging, since it includes the determination of the morphology, porosity, transport, and mechanical properties of already formed membranes.

To optimize the parameters of the membrane formation by the NIPS method, one resorts to the study of the solution itself: determination of the viscosity, construction of ternary phase diagrams, and study of the kinetics of the solution precipitation [8]. The latter is investigated using the method of a polymer deposition in a layer of an infinite thickness (“drop” method), carried out between a microscope slide and a cover glass. This approach allows visualizing the structure of the polymer layer and determining the deposition rate as the velocity of the deposition front [8,9,10]; it also makes it possible to determine the mechanism of transport of the precipitant into the volume of the polymer solution and to qualitatively evaluate the effect of the type of the polymer, its concentration, the selected solvent, and precipitant on the kinetics of the solution precipitation. Unfortunately, this method cannot adequately imitate the process of membrane formation, since it does not provide an understanding of the actual time of a polymeric membrane formation with a given thickness (<500 μm) and its morphology. The concentration profiles of the precipitant and solvent are also blurred and do not correspond to the concentration profiles in the thin film of the polymer solution. Therefore, the “drop” method allows only the comparison of the relative deposition rates of two solutions and does not allow the determination of the deposition time of a polymer film with a given thickness. Moreover, the structure of a real membrane can differ significantly from the structure formed during the deposition of a polymer layer of infinite thickness. These differences arise due to the dilution of the precipitant with a solvent and differences in the concentration profiles of the precipitant in the volume of the polymer solution [9].

The goal of this work was the development of a prompt technique to evaluate the kinetics of the deposition of polymers in a thin layer (<500 μm) for the prediction of the porous structure of thin membranes. As a model polymer we chose polyamic acid, which is the product of the reaction between resorcinol dianhydride R (1,3-bis-(3,3′,4,4′-dicarboxyphenoxy) benzene) and four nuclear diamine BAPB (4,4′-bis(4”-aminophenoxy) biphenyl). The thermoplastic polyimide R-BAPB prepared during the imidization of PAA (see Figure 1) demonstrates superior mechanical characteristics and is capable of controlled crystallization and recrystallization, as was previously exemplified by films, coatings, and binders for composite materials [5,11,12,13]. The combination of physicochemical and operational properties of PI R-BAPB, comparable to or exceeding the properties of commercial PI Kapton, Matrimid 5218, P84, ULTEM, makes it a promising material for membranes that are resistant to aggressive solutions, that can perform at elevated temperatures, and can be used in various areas, including organic solvents filtration [14].

## 2. Materials and Methods

### 2.1. List of Symbols and Acronyms

LLDP—liquid–liquid displacement porometry;NIPS—non–solvent induced phase separation;NMP—N–methyl–2–pyrrolidone;PAA—polyamic acid; PI—polyimide;PSD—pore size distribution;SEM—scanning electron microscopy;*d*—total thickness of the polymer layer, µm;*t*—time, s;*v*—deposition rate in limited layer, μm/s;*D*—diameter, nm;*γ*—interfacial tension, 10^−3^ N/m;*θ*—contact angle, °;*J*_1_—flux of the displacing liquid in the presence of the wetting liquid, L/m^2^ h;*J*_2_—flux the displacing liquid only, L/m^2^ h;Δ*p*—differential pressure, bar;*η*—viscosity, mPa∙s;*P*—permeance, kg/m^2^ h bar;*C*—concentration, wt.%.

### 2.2. Materials

The monomers used were 1,3-bis-(3,3′,4,4′-dicarboxyphenoxy) benzene dianhydride (dianhydride R), melting point T_m_ = 163–165 °C (OOO TekhKhimProm, Republic of Tatarstan, Russia), dried at 140 °C in vacuo; 4,4′-bis(4”-aminophenoxy) biphenyl (BAPB), T_m_ = 198–199 °C, (VWR International, Atlanta, GA, USA), dried at 140 °C in vacuo. Phthalic anhydride, T_m_ = 131–134 °C (Sigma-Aldrich Co. LLC, St. Louis, MO, USA) was chosen to limit chain growth during polycondensation. N-methyl-2-pyrrolidone (Sigma-Aldrich Co. LLC, St. Louis, MO, USA), used as a solvent, was specially prepared by drying (over CaH_2_) and distilling; boiling point T_b_ = 202 °C, density d_4_^20^ 1.024 g cm^−3^, refractive index n_D_^20^ 1.4684. Absolute ethanol was used as a non-solvent introduced into the casting solution, and distilled water was used as a precipitant.

### 2.3. PAA Synthesis

The preparation of a PAA solution for membrane formation was carried out per the procedure described in [13]. PAA was synthesized by polycondensation of 1,3-bis-(3,3′,4,4′-dicarboxyphenoxy) benzene dianhydride and 4,4′-bis(4”-aminophenoxy) biphenyl in N-methyl-2-pyrrolidone (NMP). The equimolar ratio of the reagents was strictly observed during the synthesis. PAA was stirred for 4 h in an inert argon atmosphere at room temperature. The initial concentration of PAA at the polycondensation stage was calculated to produce a solution with 20 wt.% of the prepolymer in the amide solvent. As it was noted in the literature [15], the process of preparation of PAA with a high molecular weight is sensitive to such parameters as the monomers ratio, the concentration of the reactants solution, and the reaction temperature. The dependence of the molecular weight on these parameters demonstrates an extremum [15]. Optimization of the conditions for PAA synthesis was performed at fixed temperature (293 K), fixed molar ratio of monomers (1.00:1.00); we varied the concentration of the solution of the reacting substances. It was found that the maximum intrinsic viscosity of PAA R-BAPB corresponds to the 20 wt.% concentration of the reagents in the solution. However, the viscosities of PAA solutions were quite high; in addition, they contained microgels in their composition which are unwanted. When filtered, the solutions had to be heated up to 60–80 °C, which, in turn, negatively affected the deformation-strength characteristics of the resulting fibers. Therefore, the solutions were diluted to 18 wt.%, which made it possible to filter them at room temperature. It was also found that the additional dilution of the already formed polyamic acid in solution does not have a strong effect on the molecular weight of the produced PAA [15].The PAA solutions were filtered and then degassed in a vacuum drying chamber for 4 h.

### 2.4. Ternary Phase Diagrams

The phase diagrams of the PAA–NMP systems at T = 25 °C were determined by means of cloud point method reported elsewhere. Water and ethanol were used as titrants. PAA solutions of various concentrations were prepared in sealed containers and titrated by water or EtOH under constant stirring at 25 °C. The titrant was added dropwise with a syringe until the solution became turbid. The cloud point was marked when the turbidity of the solution did not disappear for 24 h [16,17,18].

### 2.5. Determination of Precipitation Rate

Phase separation of the PAA-NMP solution upon contact with a precipitant was studied by two methods. The first method was performed with an “infinite” layer of polymer solution and consisted of placing a drop of polymer solution (~10 µL) between a microscope slide and a cover glass. Thus, a layer of about 2 cm in diameter was formed between the glasses. The precipitant was added dropwise using a Pasteur pipette so that it flowed into the gap between the glasses and was in contact with the polymer solution. The process of phase separation of the PAA solution was observed normally to the cover glass using a Micromed R-1 optical microscope and was recorded on a digital camera (HiROCAM MA88, Premiere, Tonawanda, NY, USA). 

A new technique for measuring the deposition rate in a “limited” layer of polymer solution was developed in the course of this work. It allows simulating the formation of a flat polymer membrane of a given thickness and visualizing the process of pore formation in an asymmetric membrane. Figure 2 represents a schematic of this technique. By gluing two cover glasses with double-sided tape, a rectangular channel with a depth (d) of 300–400 µm was formed, open to the atmosphere on one side. The channel thickness was equal to the thickness of a double-sided tape (100–105 µm). The channel was then filled with the PAA-NMP solution, and the whole assembly was fixed on a microscope slide; this slide was placed horizontally, normal to the optical axis of the microscope. A precipitant was added to the polymer solution using a Pasteur pipette from the side open to the atmosphere, and the process of phase separation of the PAA solution was observed with a microscope and recorded on camera.

The kinetics of polymer deposition was evaluated using the deposition rate of a layer of a polymer solution of a given thickness. It was calculated as the ratio of the total thickness of the polymer layer (*d*, µm) to the time of its deposition (*t*, s) (Figure 3).
(1)$$v=\frac{d}{t}$$

The deposition rate was averaged over 5 measurements for each polymer solution.

The process of deposition of 18 wt.% and 12 wt.% PAA solutions with different EtOH content (0–35 wt.%), acting as a co-solvent, was also investigated.

### 2.6. Preparation of Flat Asymmetric Membranes Based on PAA

The prepared PAA casting solutions were applied to glass using a steel cylinder with variable diameter to spread the solution, after which the solution was immersed in a bath with a precipitant (distilled water). The precipitated membrane was washed in isobutyl alcohol and then placed in a container with pure isobutyl alcohol, where it was stored to avoid contact of the polymer with water vapor.

### 2.7. Scanning Electron Microscopy

The structure and morphology of PAA membranes were studied on a Tabletop TM 3030 Plus electron microscope (Hitachi, Krefeld, Germany). The samples were transversely cleaved in liquid nitrogen to analyze the structure of the membrane. A thin conductive gold layer was deposited on the surface before placing the samples inside the microscope chamber. The accelerating voltage was 3–5 kV.

### 2.8. Study of Pore Size and Membrane Permeance

The pore size distribution (PSD) was measured by a liquid-liquid displacement porometry (LLDP) [19] using porometer POROLIQ 1000 ML (Porometer, Berlin, Germany). The measurements were carried out at 25 °C by using a pair of immiscible liquids obtained from the demixing of a mixture of iso-butanol and water (1/1, *v*/*v*). The alcohol-rich phase was used as the wetting liquid and the water-rich phase was used as a displacing liquid. 5 coupons with a diameter of 2.5 cm were cut from every membrane and were placed into the baker with wetting liquid for at least 2 h at 20 °C before the testing. The results were averaged for all 5 investigated samples. The operating principle is based on the measurement of the flux of the displacing liquid corresponding to an equilibrium pressure. The displacement of the wetting liquid was carried out by stepwise increasing the transmembrane pressure while monitoring the flux through the membrane for at least 180 s at each applied pressure. The measurement was stopped after reaching a linear dependence of the flux on pressure, which indicates complete displacement of the wetting liquid. The diameter (D) of the smallest open pore is related to the pressure at which the displacing liquid penetrates the pore via the Young-Laplace equation:(2)$$D=\frac{4\gamma \cos\theta }{\Delta p}$$ where *γ* the interfacial tension between the two liquids, *θ* the contact angle between the membrane and wetting liquid (complete wetting is assumed, i.e., cos*θ* = 1), Δ*p* is the transmembrane pressure. Interfacial tension *γ* for the mixture of iso-butanol and water is 1.9·10^−3^ N/m at 25 °C.

This method is based on the analysis of the dependence of the displacing liquid flow (iso-butanol-saturated water) on the transmembrane pressure. The initial value for calculating the pore size distribution (PSD) is the ratio of the flows of the displacing liquid in the presence of a wetting liquid (water-saturated iso-butanol) (denoted by *J*_1_) and without it (denoted by *J*_2_). The integration should be carried out from the diameter of the largest pore in the membrane (corresponds to the pressure at which the minimal reliably measured liquid flow is observed) to the diameter of the smallest pore (corresponds to the pressure at which a linear dependence of the flow on the transmembrane pressure is established). The distribution function is equal to zero outside this range. Therefore, for convenience of calculation, it is integrated in the range of pore diameters from 0 to infinity. The average pore size was calculated in the same way in [20].
(3)$$F(\Delta p)=J_{2}(\Delta p)/J_{1}(\Delta p)$$

Function *F*(∆*p*) can be converted to the function *F(D)* using Equation (4). The function *F(D)* represents the fraction of open pores with a diameter greater than or equal to *D*. In other words, it is nothing more than a cumulative distribution function. The differential function of the distribution of flow-through pores, *f(D)*, i.e., the PSD, equals the (negative) derivative of *F(D):*(4)$$f(D)=-\frac{dF(D)}{dD}$$

Note that the function *f(D)* is called “the permeance probability density function” [21] or “the flow-weighted” PSD [22]. The average pore diameter was calculated in the usual way:(5)$$\langle D\rangle =\int_{0}^{\infty }Df(D)dD$$The corrected sample standard deviation from the mean value (Ȳ) in the given series of measurements was taken as the measurement error (Δ*Y*):(6)$$\Delta Y=\sqrt{\sum_{n-1}^{n}\frac{(Y_{n}-\bar{Y{)}^{2}}}{n-1}}$$

## 3. Results

### 3.1. Phase Inversion of Polymer/Solvent Solution

The majority of asymmetric flat-sheet or hollow fibers are formed by the phase separation of a polymer solution induced by the contact with non-solvent. The diffusion of solvent and non-solvent molecules in the opposite directions, from and to a polymer solution, respectively, drives the system further out of the equilibrium resulting in the polymer precipitation. This phase inversion process can be monitored in real-time and characterized with optical microscope. However, it is important to notice that the thickness of polymer solution during the formation of asymmetric membranes does not typically exceed 100–300 µm, whereas the one or few drops of the same solution placed between two parallel optical cover glasses can form a liquid phase layer with a thickness of up to several millimeters, which can be already considered as a layer of nearly infinite thickness with regards to the local processes taking place on the interface between two phases. To illustrate the role of a thickness polymeric solution on the phase inversion process, PAA was coagulated from its 12 wt.% solution in NMP having a thickness of about 2 cm (“infinite” layer) and 300 µm (“limited” layer). Figure 4 illustrates the first 12 s of PAA precipitation in the top part (300 µm) of “infinite” layer and the “limited” layer 300 µm of thickness. The polymeric solution was placed between two cover optical glasses with a gap width of 100 µm, and then the water was dripped onto the edge of the glass.

In the beginning, a thin skin layer appeared at the interface between PAA solution and water. The porous sub-surface layer was formed within the first 4–6 s, and further polymer precipitation is mainly attributed to the formation of finger-like macrovoids growing normally to the interface. It can be seen that the polymer was precipitated faster by forming the well-established porous structure once the process took place in the “limited” layer of the polymer solution (see right images on Figure 4). Moreover, the polymer film is characterized by a thicker skin layer (~100 µm) and a more structured back-layer in contrast to the film formed in the “infinite” regime. The difference in the behavior can be attributed to the presence of the bulk polymer solution in contact with the water for the latter method. In such a case, the continuous diffusion of the solvent molecules (NMP) from the bulk to the interface, and the capability for the non-solvent molecules (water) diffuse further from the interface into the bulk drive the polymer solution slower to a two-phase system. Once the polymer solution layer is limited on one side by the solid surface, faster accumulation of the non-solvent molecules was observed, which was characterized by the greater average speed of precipitation front (34.7 vs. 29.8 µm/s).

### 3.2. Phase Inversion of Polymer/Solvent/Non-solvent Solution

The preliminary results revealed that the viscosity of polymer solution plays a role in the phase inversion process since the increase of PAA concentration in NMP from 12 (*η* = 10,700 mPa·s) up to 18 wt.% (*η* = 180,000 mPa·s) resulted in a more pronounced difference in the average speed of precipitation front for “limited” (8.3 µm/s) and “infinite” (1.7 µm/s) layer of polymer solution. Further study with 18 wt.% PAA solution was not carried out due to the extremely high viscosity of polymer solution towards membrane formation.

It is known that polyamic acids are characterized by strong inter-molecular interactions resulted in high viscosity of polymer solutions [23], and the presence of protic solvents like water or alcohol can disrupt such interactions by replacement of one carboxylic group of PAA in these hydrogen bonds. It was reported [8] that introduction of ethanol to casting solutions based on PMDA-ODA PAA allowed to reduce their viscosity and to fabricate the spin hollow fiber membranes with the desired mechanical properties. Figure 5 shows the phase diagram for PAA, NMP, and water or ethanol as non-solvent. It can be seen in Figure 5a that the polymer precipitation starts at the water content of 12.8–18.6 wt.% with regards to PAA concentration. Moreover, the hydrolysis of PAA takes place once the water is present in the polymer solution. However, ethanol can be considered a very promising alternative because the polymer solutions remain stable at an ethanol concentration of up to 40.5–44.3 wt.% (Figure 5b). In other words, up to 50 wt.% of NMP solvent in the casting solution can be replaced by ethanol, which can have a renewable origin.

As can be seen from Figure 6, the partial replacement of NMP with ethanol led to a decrease in the viscosity of PAA solution; it should be pointed out that PAA concentration in all solutions remained constant as 12 wt.%. The increase of ethanol concentration up to 20 wt.% was followed by a nearly linear drop in the polymer solution viscosity from 10,700 to 4000 mPa∙s. Farther increase of the alcohol content up to 35 wt.% did not significantly change the viscosity (drop from 4000 to 2700 mPa∙s). As discussed above, the drop in viscosity can be associated either with the disruption of interchain coupling due to hydrogen bonding or the change of thermodynamic properties of the polymer solution since ethanol acts as a non-solvent.

Figure 7 summarizes the results of the precipitation of polymer film from PAA/NMP/EtOH solutions having an “infinite” and “limited” layer by using water as non-solvent (left and right images, respectively). The same polymer solution was used to cast a flat-sheet asymmetric membrane, and their cross-sections visualized by SEM are also represented in Figure 7 (images in the center). It can be seen that the replacement of NMP by ethanol in the polymer solution resulted in noticeable changes of the membrane morphology; for instance, the size of macrovoids in the support layer was decreased with the increase of alcohol concentration and their shape became elongated finger-like at an alcohol content of 35 wt.%. The walls of formed macrovoids had a spongy porous structure, and its proportion increased with the further replacement of NMP by ethanol [8]. It can be concluded from Figure 7 that there is a good agreement in the morphology of the porous structure of polymer film formed by the precipitation from a thin film of a polymer solution and membrane cast from the same solution (see corresponded images in the center and right). Meanwhile, the morphology of the porous structure of polymer films formed in “infinite” layer on the interface of two bulk phases does not fully correspond to the membrane structure.

It was earlier shown in Figure 6 that the replacement of NMP by ethanol reduced PAA solution viscosity, and the faster kinetics of precipitation can be expected for a less viscous solution. Figure 8 shows the position of precipitation front from the interface during the first 7 s for 12 wt.% PAA solutions containing 0, 15, 20, and 35 wt.% of ethanol. It can be seen that the addition of ethanol changed significantly the precipitation behavior of polymer solutions. Due to lower viscosity (2700–5900 mPa∙s), attributed to the partial disruption of interchain coupling of PAA by ethanol, the water molecules diffused faster in the polymer solution, which resulted in more rapid precipitation. However, when the thickness of the PAA layer was about 50–70 µm, the precipitation rate of a solution slowed down noticeably. At the same time, the 12 wt.% PAA system (10,700 mPa∙s) possessed a nearly constant precipitation rate, showing the linear progress of precipitation front over time. Such difference in the kinetics of polymer precipitation can be attributed to the structure and properties of the sub-surface layer formed in the beginning, since the further diffusion of solvent and non-solvent molecules through the interface would be limited particularly by their transport through this layer.

To get an insight into the porous structure formed during the phase inversion, the asymmetric membranes fabricated from 12 wt.% PAA solutions with different ethanol content were characterized by using the liquid-liquid displacement porometer. As can be seen from Figure 9, with an increase in alcohol concentration in casting solution from 0 up to 15 wt.%, there was a sharp decline in the average size of the transport pores from 80 down to 12 nm. A further increase in the ethanol concentration was accompanied by a slight increase in the average size of transport pores. Thus, a decrease of the polymer precipitation rate after the first few seconds of phase inversion of ethanol-based casting solutions can be associated with the formation of a low-permeance sub-surface layer having narrower pore size distribution. The formation of such a layer hindered the transport of water into the polymer solution. In the case of PAA/NMP solution, the polymer film formed from the viscous solution was characterized by bigger pores, which provided lower mass-transfer resistance to the diffusion of solvent and non-solvent molecules. As a result, the polymer precipitation rate remained nearly constant (see Figure 8 for 0 wt.%).

The text of the article was supplemented by a comparison of the morphology of the prepared membranes with the literature data. Similar structures were obtained in [8,24], where the influence of the addition of EtOH into the casting solution on the morphology of hollow fiber PAA PMDA-ODA membranes was investigated. As in this work, the authors of [8,24] observed an increase in the proportion of the walls of finger-shaped pores with ethanol concentration in the casting polymer solution.

The principle of porometry used in this study is based on the stepwise displacement of one phase (iso-butanol rich water solution) by another one (water-rich iso-butanol solution), which are immiscible and in equilibrium with each other. Therefore, it was possible to determine the permeance of the water-rich phase once all pores were displaced and open for transport. It can be seen from the left graph in Figure 10 that the permeance of the water-rich phase through different membranes was in good agreement with the average pore size presented in Figure 9. With an increase in ethanol concentration up to 15 wt.%, the average pore size decreased to 12 nm, while the liquid permeance through the membranes dropped by an order of magnitude from 207 to 16.6 kg/m^2^∙h∙bar. A further increase in the alcohol content up to 30–35 wt.% led to an increase in the average pore size from 16.5 to 27 nm, while the liquid permeance increased about threefold. It is interesting to notice that there is a certain correlation between the precipitation rate and the liquid permeance of the corresponded membranes (see the right graph in Figure 10). Indeed, both the diffusion of non-solvent/solvent molecules during the phase inversion and liquid transport at the membrane filtration would be mainly limited to the sub-surface layer having the narrowest pore structure. To sum up, the membrane morphology and transport properties can be estimated by the precipitation of a small quantity of corresponded polymer solution within the thin layer, without casting and characterization of asymmetric membranes.

## 4. Conclusions

In this work, the precipitation of a thin layer of a polymer solution was proposed to imitate the process of asymmetric membranes formation by non-solvent induced phase separation (NIPS) technique. The solution of polyamic acid (PAA) in NMP was placed in the gap of 100 μm between two parallel glasses, and then the precipitation kinetics was monitored by the optical microscope once the distilled water as non-solvent was added. The barrier between two glasses was introduced that allowed to form a thin polymer solution film (<500 μm), so-called “limited” regime; the experiments without a barrier (polymer solution layer of ~2 cm) was also carried as the reference, so-called “infinite” regime. Polymer films formed within the “limited” layer of PAA solution showed a good agreement with the morphology of asymmetric flat-sheet membranes prepared from the same solutions by NIPS method, but the polymer films formed on the interface of two bulk phases (“infinite” regime) did not fully correspond to the membrane structure.

PAA-EtOH-NMP phase diagram showed that up to 50% of NMP solvent in the casting solution can be replaced by ethanol, which can have a renewable origin. With the addition of 35 wt.% of ethanol, the viscosity of 12 wt.% PAA solution can be reduced from 10,700 to 2700 mPa∙s, which can be attributed to the partial disruption of interchain coupling as a result of hydrogen bonding. It was shown that the presence of alcohol significantly changed the behavior of precipitation kinetics, the morphology of formed films, and asymmetric membranes. There is a certain relationship between the average precipitation rate within the “limited” polymer solution layer and the permeance of a liquid through the corresponding asymmetric membranes.

To summarize, the precipitation of a polymer solution within the thin layer can be considered a prompt technique and a powerful tool for the fast screening and optimization of the complex composition of casting solutions, using its small quantity. Furthermore, the prediction of membrane morphology can be done without casting the membrane, required post-treatment procedures, and SEM analysis.

## Figures and Tables

**Figure 1 polymers-12-02785-f001:**
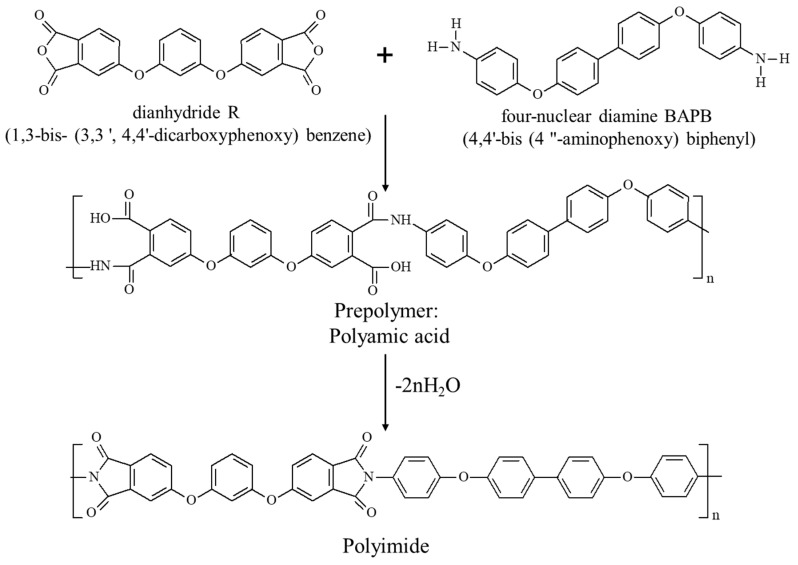
Synthesis of PI R-BAPB.

**Figure 2 polymers-12-02785-f002:**
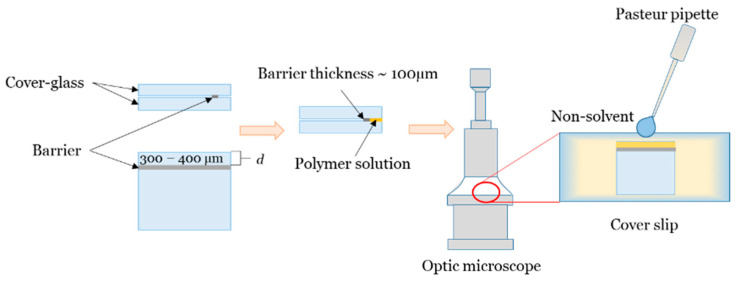
Scheme of a new technique for determination of the deposition rate in polymer solutions, which imitates the formation of a flat polymer membrane.

**Figure 3 polymers-12-02785-f003:**
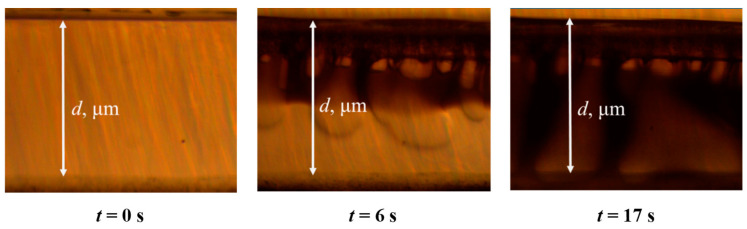
Photomicrographs captured in the study of the deposition kinetics by a new method.

**Figure 4 polymers-12-02785-f004:**
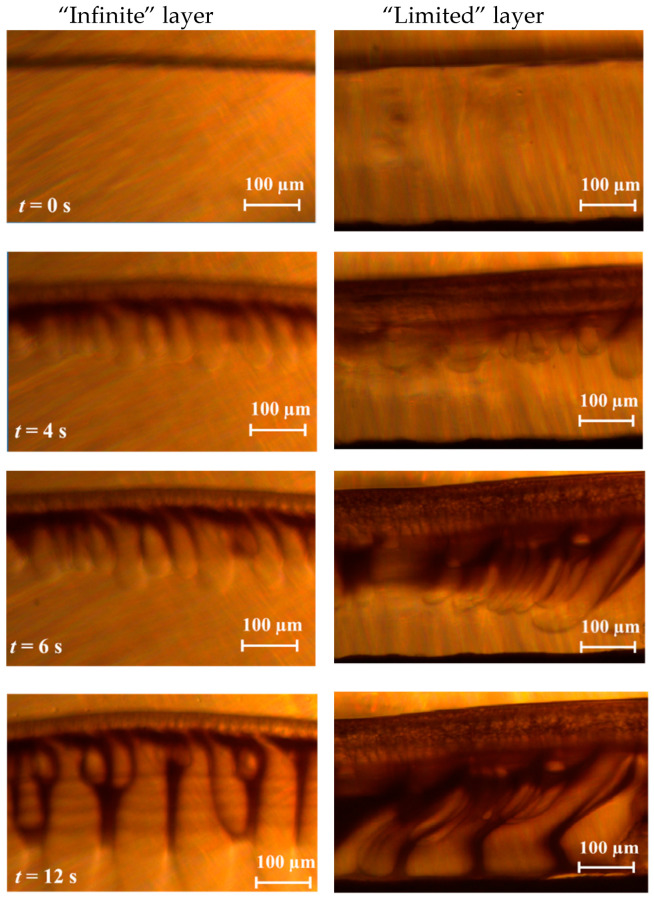
The precipitation kinetics of 12 wt.% solution of PAA in NMP upon contact with water; the thickness of polymer solution was about 2 cm (left) and 300 µm (right).

**Figure 5 polymers-12-02785-f005:**
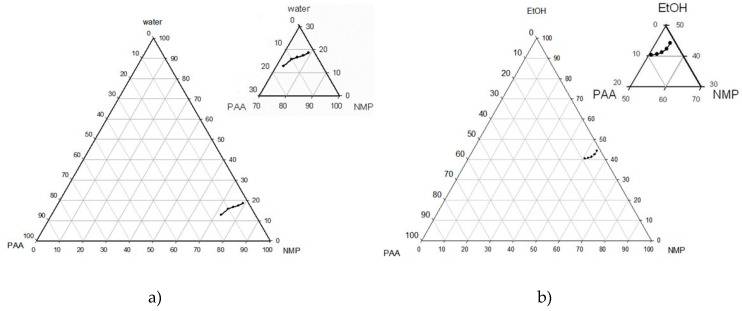
Phase diagram for PAA-NMP-H_2_O (**a**) and PAA-NMP-EtOH (**b**).

**Figure 6 polymers-12-02785-f006:**
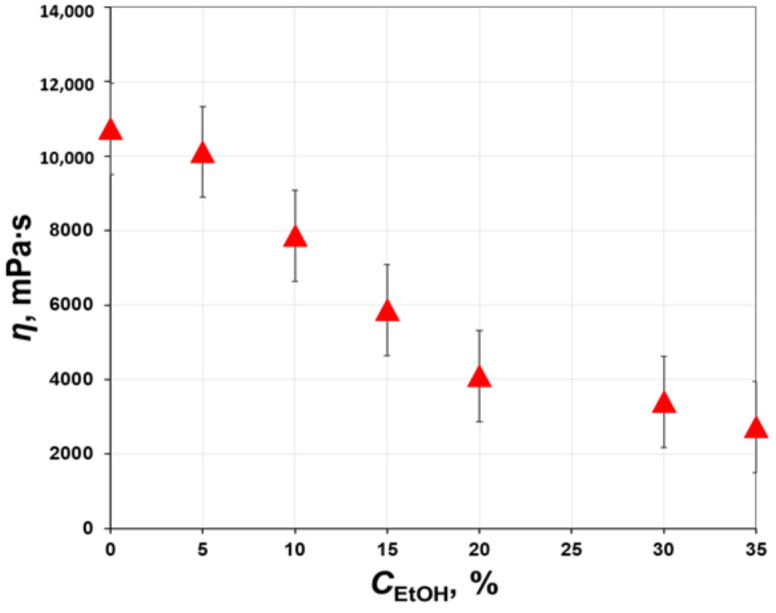
The effect of ethanol content on the viscosity of 12 wt.% PAA solution in NMP.

**Figure 7 polymers-12-02785-f007:**
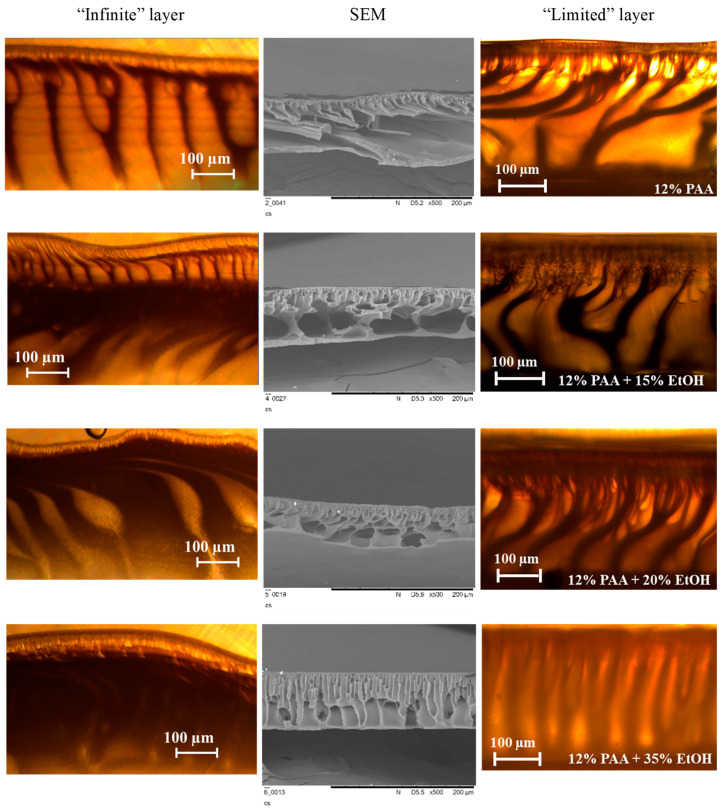
Effect of ethanol concentration on the PAA film formed from “infinite” (**left**) and “limited” (**right**) layer of a polymer solution, and SEM cross-section of the membrane (**center**).

**Figure 8 polymers-12-02785-f008:**
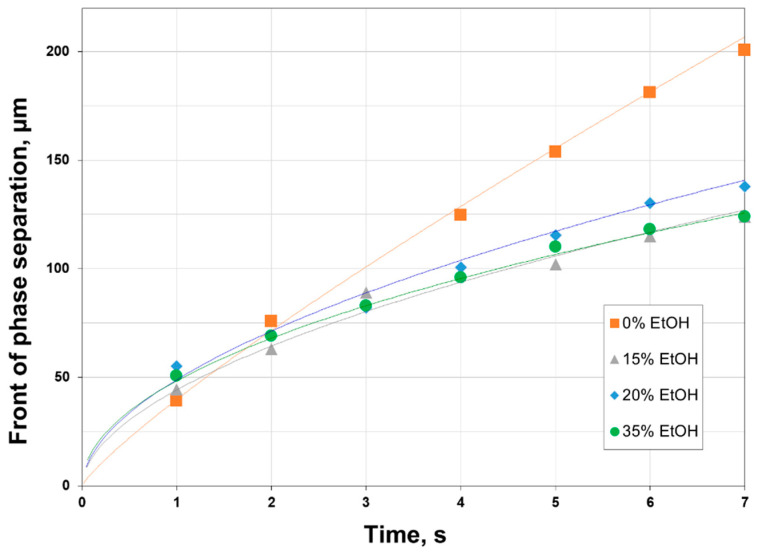
Change of the phase separation front as a function of time (the trend lines are provided as eye-guidance).

**Figure 9 polymers-12-02785-f009:**
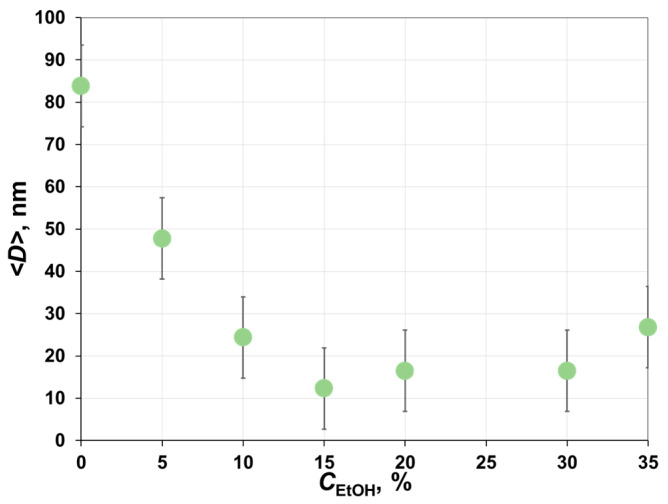
Effect of ethanol concentration in casting solution on the average diameter of transport pores of membranes.

**Figure 10 polymers-12-02785-f010:**
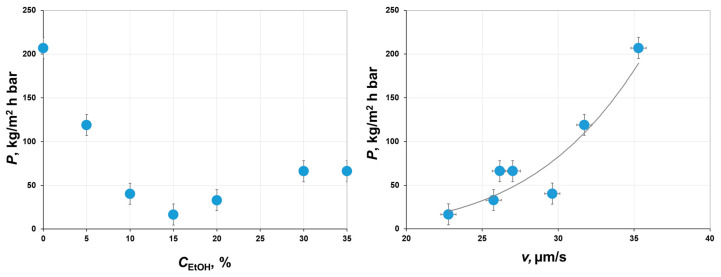
Effect of ethanol concentration in casting solution C_EtOH_ (left) and the average precipitation rate v (right) on permeance of water-rich iso-butanol solution through the corresponded membranes.

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
