# Peer review of "Phase Separation within a Thin Layer of Polymer Solution as Prompt Technique to Predict Membrane Morphology and Transport Properties"

_polymers, 2020, doi:10.3390/polym12122785_

Round 1

Reviewer 1 Report

This manuscript concerns the investigation of polymeric membrane morphology prediction using two methods (ILS and LLS) during non-solvent induced phase separation. This manuscript is easy to read and authors introduced new technique in investigating asymmetric membrane formation during its fabrication. However, several concerns need to be amended and address before the manuscript can be accepted for publication. Here are some comments:

  1. Page 3, line 96 - The authors mentioned on the produced solution with 20 wt.%, but later due to high viscosity of 20 wt.% solution, the authors change to 18 wt. % solution. Why do authors decide with 20 wt. % at the first place? it the number in the range of author's target application? the justification should be included in the introduction section as one of the result of author's literature review.
  2. Section 2.4 - equation 1 - there are no data on the deposition rate in the RnD section. kindly please verify if the eq. 1 is used in this manuscript.
  3. Its there any data presented and result discussed from the first method (ILS)? 
  4. Page 7 - Figure 4 - Authors mentioned on the different in thickness of the polymeric solution, which is 2 cm (left figures) and 300 um (right figures). But however, from my point of view, both right and left seem to be similar figures and i couldn't differentiate it. Furthermore, the authors was not mentioning on the 2cm thickness in the methodology section.
  5. Page 9, line 257 - the authors should compare with several previous literatures on the polymer morphology obtained in this manuscript to validate the author's observation result.
  6. Page 12, Line 300 - the authors compare the permeance with its own pore size result in this line. However, it would be good if the permeance result obtained can be compare with the previous literatures with similar application.

Author Response

  1. Page 3, line 96 - The authors mentioned on the produced solution with 20 wt.%, but later due to high viscosity of 20 wt.% solution, the authors change to 18 wt. % solution. Why do authors decide with 20 wt. % at the first place? it the number in the range of author's target application? the justification should be included in the introduction section as one of the result of author's literature review.

Answer: Polyamic-acid R-BAPB was produced by polycondensation of stoichiometric amounts of diamine BAPB (4,4'-bis(4"-aminophenoxy) biphenyl) with resorcinol dianhydride R (1,3-bis-(3,3',4,4'-dicarboxyphenoxy) benzene) in NMP at 25 °C. As it was noted in the literature [15], the process of preparation of PAA with a high molecular weight is sensitive to such parameters as the monomers ratio, the concentration of the reactants solution, the reaction temperature; the dependence of the molecular weight on these parameters demonstrates an extremum [15]. Optimization of the conditions for PAA synthesis was performed at fixed temperature (293 K), fixed molar ratio of monomers (1.00:1.00); we varied the concentration of the solution of the reacting substances. It was found that the maximum intrinsic viscosity of PAA R-BAPB corresponds to the 20 wt. % concentration of the reagents in the solution. However, the viscosities of PAA solutions were quite high; in addition, they contained microgels in their composition which are unwanted. When filtered, the solutions had to be heated up to 60-80 °C, which, in turn, negatively affected the deformation-strength characteristics of the resulting fibers; therefore the solutions were diluted to 18 wt. %, which made it possible to filter them at room temperature. It was also found that the additional dilution of the already formed polyamic acid in solution does not have a strong effect on the molecular weight of the produced PAA [15].

  1. Silinskaya, I.G., Svetlichnyi, V.M., Kalinina, N.A., Didenko, A.L., Filippov, A.P., Kudryavtsev, V.V. Molecular characteristics and solution behavior of prepolymers of several polyimides: effect of synthesis conditions. Polymer Science. Series A. 2006, 48, 787-792. https://doi.org/10.1134/S0965545X06080037

Answer added on page 4 section 2.3.

  1. Section 2.4 - equation 1 - there are no data on the deposition rate in the RnD section. kindly please verify if the eq. 1 is used in this manuscript.

Answer: Equation 1 is used to calculate the deposition rate of a polymer solution in the infinite layer of polymer solution method. The data are shown in section 3.1 on page 8, line 255-256 and section 3.2 page 9, line 261.

  1. Its there any data presented and result discussed from the first method (ILS)?

Answer: Perhaps, the original text did not clearly describe the comparison of the two methods for studying the kinetics of deposition. The photos obtained with the first investigated method (“infinite” layer) are presented in section 3.1 in Figure 4 (left column). The discussion to the figure goes below and includes a comparison of the process of phase separation of a 12 wt. % PAA solution in NMP, investigated by two methods (“infinite” layer and “limited” layer). Additional explanations are included in the text on page 7 lines 233-237.

Photos obtained using the “infinite” layer method are shown in Figure 7 (left column). The discussion to the figure includes the analysis of the morphology of samples formed in the infinite and limited layers upon the addition of various amounts of EtOH.

  1. Page 7 - Figure 4 - Authors mentioned on the different in thickness of the polymeric solution, which is 2 cm (left figures) and 300 um (right figures). But however, from my point of view, both right and left seem to be similar figures and i couldn't differentiate it. Furthermore, the authors was not mentioning on the 2cm thickness in the methodology section.

Answer: Thank you for your comment. Information about the layer thickness (2 cm) in the “infinite” layer method is added to the experimental part on page 4, line 142.

Figure 4 (left column) shows not the entire 2 cm layer, but 300 µm, which corresponds to the layer thickness in the “limited” layer method. This was done in order to compare the speed of the reaction front for an equal period of time in the “infinite” and “limited” layer.

  1. Page 9, line 257 - the authors should compare with several previous literatures on the polymer morphology obtained in this manuscript to validate the author's observation result

Answer: Thanks for the constructive comment; the text of the article was supplemented by a comparison of the morphology of the prepared membranes with the literature data. Similar structures were obtained in [Li, Y.; Cao, B.; Li, P. Fabrication of PMDA-ODA hollow fibers with regular cross-section morphologies and study on the formation mechanism. J. Membr. Sci. 2017, 544, 1-11; H. Lee, J. Won, H. Park, H. Lee, Y. Kang, Effect of poly(amic acid) imidication on solution characteristics and membrane morphology, J. Membr. Sci. 178 (2000) 35–41], where the influence of the addition of EtOH into the casting solution on the morphology of hollow fiber PAA PMDA-ODA membranes was investigated. As in this work, the authors of [Li, Y.; Cao, B.; Li, P. Fabrication of PMDA-ODA hollow fibers with regular cross-section morphologies and study on the formation mechanism. J. Membr. Sci. 2017, 544, 1-11; H. Lee, J. Won, H. Park, H. Lee, Y. Kang, Effect of poly(amic acid) imidication on solution characteristics and membrane morphology, J. Membr. Sci. 178 (2000) 35–41] observed an increase in the proportion of the walls of finger-shaped pores with ethanol concentration in the casting polymer solution.

Answer added on page 12 section 3.2, lines 335 – 340.

  1. Page 12, Line 300 - the authors compare the permeance with its own pore size result in this line. However, it would be good if the permeance result obtained can be compare with the previous literatures with similar application.

Answer: Studies of the influence of PAA membranes formation conditions on their morphology do exist [Li, Y.; Cao, B.; Li, P. Fabrication of PMDA-ODA hollow fibers with regular cross-section morphologies and study on the formation mechanism. J. Membr. Sci. 2017, 544, 1-11]. However, the open literature doesn’t provide data on the filtration of liquids through such membranes. Also, the pore size distribution in PAA membranes has not been previously evaluated. Therefore, it is currently not possible to compare the permeance data obtained in this work with the literature data.

Reviewer 2 Report

A file with possible changes is attached

Author Response

  • A list of nomenclature and acronyms would help to follow the text.

Answer: Experimental part of the article was expanded by addition of “2.1 List of symbols and acronyms.”

LLDP - liquid-liquid displacement porometry;

NIPS - non-solvent induced phase separation;

NMP - N-methyl-2-pyrrolidone;

PAA - polyamic acid;

PI - polyimides

PSD - pore size distribution;

SEM - scanning electron microscopy;

d - total thickness of the polymer layer, µm;

h - thickness of the deposition front, μm;

t - time, s;

vn-l  - deposition rate in infinite layer, μm/s;

vn - deposition rate in limited layer, μm/s;

D – diameter, nm;

γ - interfacial tension, 10-3 N/m;

θ - contact angle, o;

J1 - flux of the displacing liquid in the presence of the wetting liquid, l/m2 h;

J2 - flux the displacing liquid only, l/m2 h;

Δp - differential pressure, bar;

η - viscosity, mPa∙s;

P - permeance, kg/m2 h bar;

C - concentration, wt. %.

  • A mention to the uncertainties should be done in the text. They are nicely plotted on figures 6, 9 and 10, but nothing is explained in the text. How are they being obtained in each case?

Answer: The corrected sample standard deviation from the mean value (Ȳ) in the given series of measurements was taken as the measurement error (ΔY):

Answer added on page 7 section 2.8, lines 219 – 221.

  • On page 6, where the mathematical description of the method is explained, I miss a better definition of the variables (for instance J). The integral in (6) goes from 0 to infinite, what is the physical meaning? More in-depth explanations would be needed here.

Answer: The description of the method expanded in accordance with the recommendations of the reviewer. This method based on the analysis of the dependence of the displacing liquid flow (Isobutanol-saturated water) on the transmembrane pressure. The initial value for calculating the pore size distribution is the ratio of the flows of the displacing liquid in the presence of a wetting liquid (water-saturated Isobutanol) (denoted by J1) and without it (denoted by J2). Of course, the reviewer is absolutely correct that, from a physical point of view, it makes no sense to integrate the pore size distribution function from 0 to infinity. The integration should be carried out from the diameter of the largest pore in the membrane (corresponds to the pressure at which the minimal reliably measured liquid flow is observed) to the diameter of the smallest pore (corresponds to the pressure at which a linear dependence of the flow on the transmembrane pressure is established). The distribution function is equal to zero outside this range; therefore, for convenience of calculation, it is integrated in the range of pore diameters from 0 to infinity. The average pore size was calculated in the same way in [Antón, E., Calvo, J. I., Álvarez, J. R., Hernández, A., & Luque, S. (2014). Fitting approach to liquid–liquid displacement porometry based on the log-normal pore size distribution. Journal of membrane science, 470, 219-228.], Equation 19.

Answer added on page 6 section 2.8, lines 201 – 210.

  • The unit used in the abstract and on page 12, together with figure 10: kg/m2•h•bar, I am sorry, but I do not follow it. There is also a mN/m in page 6, that I suggest changing it to 1.9•10-3 N/m. If the first m means mili-Newtons.

Answer: Thank you for your comment. On page 13, line 351, the exact permeances for PAA membranes are given as shown in Figure 10 (Left). On page 6, line 200, the mN/m unit was changed to 1.9 10-3 N/m, in accordance with the reviewer's suggestion.

  • My other main concern is on the graphs lay-out. I am referring to figures 6, 8, 9 and 10. I would suggest some uniform template (figure 8 is totally different, but the others have substantial formatting differences, also). The use of intermediate or gridlines would help to better understand the results.

Answer: A uniform template was used for these figures, and gridlines have been added to the figures for better readability.

  • A final global reading and English Grammar correction are also recommended.

Answer: The article was revised and grammatical errors corrected, using “American English” template from Grammarly software.

Particular corrections: Thanks a lot for the found mistakes.

  • – from 1.6 up to 207…” . Instead of a dash “–“, I suggest to use parenthesis there.

Corrected, page 1, lines 23-24.

  • Introduction, line 43, instead of “laborious” I would recommend challenging.

The word “laborious” was replaces by “challenging” on page 1, line 43.

3) Equation 3, could be better as:

Equation 3 was corrected.

  • On page 6, apart from the mathematical model description, that is quite weak (as already mentioned), what is the “optical method” (line 187). 2

In this case, by the optical method we mean the study of the kinetics of the deposition of a polymer solution using an optical microscope. Fixed on optical microscope on page 6 line 189.

  • On page 7, line 214, there is again a dash “–“, to mention “… – vs 29.8 μm /s)”. The same happens in the foot of figure 4, and probably somewhere else (in the abstract it was already pin pointed).

Corrected on page 7, line 228.

  • The conclusions mention the H-bond and it was not so much commented in the previous text. Also, the concepts “limited” layer and “infinite” regime are not too much linked with previous text. At least, in that shape. Probably, it would help the reader to follow the statements in the conclusions.

We replaced “H-bond” on “hydrogen bond” on page 14, line 379. The experimental part introduced the terms “limited” layer and “infinite” layer on page 4, line 140 and page 5, line 149.

  • As a minor correction, in the conclusions, lines 319 and 322, there is μM instead of μm

Corrected, lines 367 and 370.
